

# Age-dependent differences in iris colouration of passerines during autumn migration in Central Europe

Michał Polakowski[1], Krzysztof Stępniewski[2], Joanna Śliwa-Dominiak[1] and Magdalena Remisiewicz[2]

[1] Institute of Biology, University of Szczecin, Szczecin, Poland
[2] Bird Migration Research Station, Faculty of Biology, University of Gdańsk, Gdańsk, Poland

## ABSTRACT

Avian eye colour changes with age, but many aspects of this transition are still insufficiently understood. We examined if an individual's sex, age, species and body condition are related to the iris colour in common migratory passerines during their autumn passage through Central Europe. A total of 1,399 individuals from nine numerous species were ringed and examined in late autumn in northern Poland. Each individual was sexed by plumage (if possible) and assigned to one of three classes of the iris colour—typical for immatures, typical for adults and intermediate. We found that the iris was typical in 97.7% cases of immatures and in 75.8% cases of adults and this difference was significant. Species, sex and body mass index (BMI) had no significant influence on the iris colour. We show that iris colour in passerines in late autumn is strongly age-dependent and thus can serve as a reliable feature for ageing in field studies, especially in species difficult to age by plumage.

## INTRODUCTION

The colour of bird's eyes is an important, sometimes even striking or diagnostic, part of overall colouration. The source of the colour is pigments located in the iris, including melanines, carotenoids, pteridines and purines (*Bond, 1919*; *Oliphant, 1987*; *Hudon & Muir, 1996*; *Prum, 1999*). Contrary to humans, the sclera of a bird's eye is not visible, hence the eye colour is the colour of the iris (*Davidson, Thornton & Clayton, 2017*). Although most birds have dark eyes, irises range from red and blue through yellow to white. The colour of birds' eyes might be related to size (*Worthy, 1978*), feeding strategy (*Craig & Hulley, 2004*), communication between individuals (*Davidson, Clayton & Thornton, 2014*) or breeding behaviour (*Davidson, Thornton & Clayton, 2017*). A large inter- and intraspecific variety of eye colours exist in birds, sometimes connected with sex but mostly with the age of an individual (*Newton & Marquiss, 1982*; *Sweijd & Craig, 1991*; *Bortolotti, Smits & Bird, 2003*; *Wilson & Hartley, 2007*; *Nogueira & Alves, 2008*; *Mero & Zuljevic, 2015*). Immature birds usually have dark irises, which turn paler or brighter as they mature. This phenomenon has been described and used for age determination in several families, including penguins,

Corresponding author
Michał Polakowski,
michal.polakowski@usz.edu.pl

waterfowl, raptors, gulls and some passerines (*Trauger, 1974*; *Snyder & Snyder, 1974*; *Newton & Marquiss, 1982*; *Rosenfield & Bielefeldt, 1997*; *Scholten, 1999*; *Bortolotti, Smits & Bird, 2003*). However, the source of this variety and its potential use for ageing more extensively, as well as the general role of avian eye colour, remain poorly understood (*Sweijd & Craig, 1991*; *Bortolotti, Smits & Bird, 2003*; *Craig & Hulley, 2004*; *Negro, Blazquez & Galvan, 2017*).

Most Palaearctic passerines have dark eyes (*Craig & Hulley, 2004*); the few exceptions include the Jackdaw *Corvus monedula*, Jay *Garrulus glandarius* and Bearded Reedling *Panurus biarmicus*. Changes of eye colour with age have been studied in detail in a few species of passerines, especially at Falsterbo Bird Observatory in Sweden (e.g., *Karlsson, Persson & Walinder, 1992*; *Karlsson, Persson & Walinder, 1986*; *Brensing, 1985*; *Karlsson, Persson & Walinder, 1988*; *Gargallo, 1992*; *Karlsson, Persson & Walinder, 1993*; *King & Muddeman, 1995*; *Wilson & Hartley, 2007*; *Mero & Zuljevic, 2015*). Handbooks of bird identification in hand describe iris colour as a generally reliable supporting feature for ageing passerines, along with plumage characters and skull ossification (*Svensson, 1992*; *Demongin, 2016*), but details are given mostly for species with reddish eyes, such as Dunnock *Prunella modularis* and Crested Tit *Lophophanes cristatus* (*Svensson, 1992*), for those whose adults have clearly pale irises, such as some *Sylvia* and *Acrocephalus* species (e.g., Common Whitethroat *S. communis* and Eurasian Reed Warbler *A. scirpaceus*; *Karlsson, Persson & Walinder, 1985*; *Karlsson, Persson & Walinder, 1988*), but for only a few dark-eyed species, such as Reed Bunting *Emberiza schoeniclus* or Tree Pipit *Anthus trivialis* (*Karlsson, Persson & Walinder, 1985*; *Karlsson, Persson & Walinder, 1993*). We are unaware of any study testing the reliability of this feature as an additional means of ageing on a larger sample of dark-eyed passerines.

We aimed to determine the age-dependent iris colour of several dark-eyed passerines that migrate in large numbers through Central Europe in late autumn. We also checked the potential influence of sex and body condition on this trait, expecting the relationship to be universal and unaffected by sex or body condition, making iris colour a reliable supporting feature in the ageing of these passerines.

## MATERIALS & METHODS

Data were collected at the Bukowo-Kopań bird ringing station on the southern Baltic coast of northern Poland (54°20′15″N, 16°14′40″E). The station operates within the Operation Baltic Research and Monitoring Programme, conducted by the Bird Migration Research Station, Faculty of Biology, University of Gdańsk. Migrating passerines were captured in 60 mist nets, located mainly in broad-leaved shrubs on the edge of marshy forest and sand dunes. Nets were checked every hour from dawn till dusk. All the captured birds were ringed, sexed (if possible) and aged as first calendar-year birds (hereafter immatures) or older (adults). From each bird a set of biometric measurements was collected, including fat score, wing length (maximum chord) and mass to an accuracy of 0.1 g; for more details see (*Busse & Meissner, 2015*).

**Table 1 Numbers of individuals with different classes of iris colour in the studied species; % percentage of birds aged correctly by the iris colour.** Class 1 = grey iris, class 2 = brown iris, class 3 = iris colour intermediate between grey and brown.

| Species | Iris colour | | | | | | | | | |
|---|---|---|---|---|---|---|---|---|---|---|
| | immatures | | | | | adults | | | | |
| | class 1 | class 2 | class 3 | total | % | class 1 | class 2 | class 3 | total | % |
| Great Tit *Parus major* | 54 | 0 | 4 | 58 | 93.1 | 0 | 21 | 3 | 24 | 87.5 |
| Eurasian Blue Tit *Cyanistes caeruleus* | 25 | 0 | 2 | 27 | 92.6 | 0 | 10 | 1 | 11 | 90.9 |
| Common Chiffchaff *Phylloscopus collybita* | 33 | 0 | 0 | 33 | 100 | 0 | 5 | 0 | 5 | 100 |
| Long-tailed Tit *Aegithalos caudatus* | 92 | 0 | 4 | 96 | 95.8 | 0 | 0 | 0 | 0 | – |
| Goldcrest *Regulus regulus* | 535 | 0 | 12 | 547 | 97.8 | 0 | 58 | 18 | 76 | 76.3 |
| Eurasian Treecreeper *Certhia familiaris* | 21 | 0 | 0 | 21 | 100 | 0 | 2 | 0 | 2 | 100 |
| Winter Wren *Troglodytes troglodytes* | 64 | 0 | 0 | 64 | 100 | 0 | 0 | 2 | 2 | 0 |
| European Robin *Erithacus rubecula* | 217 | 0 | 1 | 218 | 99.5 | 1 | 35 | 20 | 56 | 62.5 |
| Common Blackbird *Turdus merula* | 116 | 1 | 3 | 120 | 96.7 | 1 | 32 | 6 | 39 | 82.1 |
| Total | 1157 | 1 | 26 | 1184 | 97.7 | 2 | 163 | 50 | 215 | 75.8 |

The iris colour was examined in nine of the most numerous species from 19th October to 6th November 2019, which provided a sufficiently large sample for analysis (Table 1). Birds were assigned to one of three classes of iris colour (Figs. 1, 2):

a) class 1: iris typical for immatures—dark grey, without any brown, contrasting only slightly with the pupil,

b) class 2: iris typical for adults—paler and contrasting with the pupil, always with a visibly warm brownish colour,

c) class 3: intermediate iris colour, pale and contrasting with the pupil (as in class 2), but greyish without an obvious brownish colour.

The apparent colour of the iris to an observer could partially result from ambient light, so colour was always determined in daylight with good visibility. To avoid observer bias, the iris colour was determined and all the ringing data were collected by one of the authors (MP), who has ringed about 100,000 birds during almost 20 years in Poland and Sweden. The birds were aged based on plumage features and skull ossification, and sexed according to plumage whenever possible (*Svensson, 1992*; *Demongin, 2016*). Robins *Erithacus rubecula,* which lack sexually dimorphic plumage, were sexed on the basis of wing
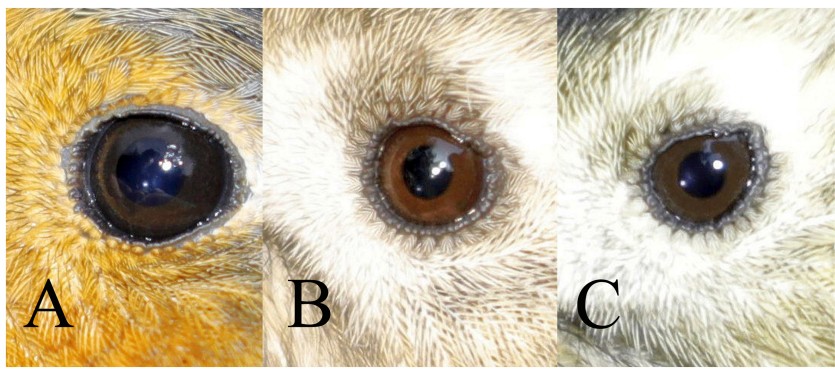

**Figure 1** **Three classes of iris colour used in this study.** A = class 1 (immature type grey iris) in Robin *Erithacus rubecula*, B = class 2 (adult type brown iris) and C = class 3 (iris intermediate between grey and brown) in Goldcrest *Regulus regulus*. Photos by Jacek Rogoziński.

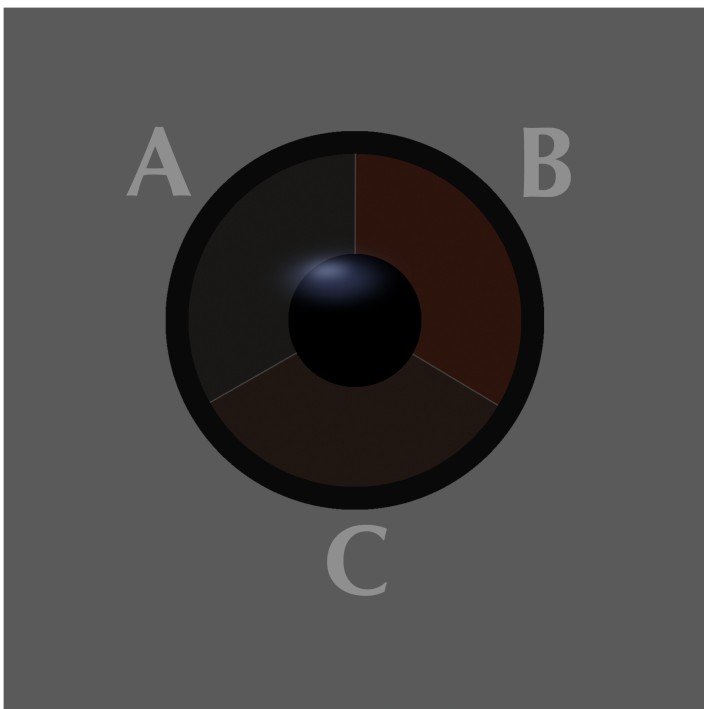

**Figure 2** **Comparison of the three classes of the avian iris colour used in the study.** A = class 1: immature type grey iris, B = class 2: adult type brown iris, C = class 3: iris intermediate between grey and brown. Drawing by Tomasz Cofta.

length, according to (*Polakowski & Jankowiak, 2012*). Using the wing length criteria of adult male >75 mm and adult female <72 mm, with immature male >74 mm and immature female <71 mm, we successfully sexed 29.6% of the Robins we caught. To avoid the bias of self-suggestion by unconsciously assigning the iris colour to the class compliant with the
age identified by plumage, we adopted two ways of identification: in 63% of individuals their age was first identified by iris colour and then checked by plumage, and in 37% the other way around. The overall numbers of birds we examined and their eye colour classes are given in Table 1. All the bird ringing was conducted with the approval of the General Directorate for Environmental Protection in Poland (DZP-WG.6401.03.2.2018.jro). Field research at Bukowo was approved by the Marine Office in Słupsk (OW-A-510/100/19jt).

We tested the relation between age and iris colour using the Chi-square ($\chi^2$) test of independence. Then we checked if age or species increased the odds of an iris colour non-compliant with age class using a generalised linear model (GLM) with binomial error distribution. The dependent variable was compliance (1 = compliant, 0 = non-compliant); the classes of iris colour were coded as: class 1 (grey) iris of immatures = 1, class 2 (brown) iris of adults = 1, class 3 (intermediate) iris of immatures = 0, class 3 (intermediate) iris of adults = 0. Adults with the immature type of iris (class 1) were considered in the model as 0 (non-compliant). We used age, species and body mass index (BMI), calculated as body mass divided by wing length, as explanatory variables. Not all individuals could be sexed, so we used a smaller data set to analyse the effect of sex, which we added to the variables in the previous model. We also added the age identification sequence (first iris then the other traits, or the other way around) as an explanatory variable to check if it had affected the results. Statistical analyses were performed using R software 3.6.1 (*R Development Core Team, 2019*), with a significance level of $p < 0.05$.

## RESULTS

We examined 1,399 individuals from nine numerous species. The iris colour was typical in 97.72% of our sample (1157 compliant - class 1, vs 27 intermediate - class 3) of immatures and in 75.81% (163 compliant - class 2, vs 52 intermediate - class 3) of adults, when all species were combined. This difference was statistically significant ($\chi^2 = 163.9, p < 0.001$). In adults the iris colour was 13.1 times more likely to be intermediate than in immatures ($\beta_{\text{Intercept}} = -4.013 \pm 0.230$ SE; $\beta_{\text{Age[adults]}} = 2.569 \pm 0.253$ SE, $\chi^2 = 107.163, p < 0.001$). Thus, 93.1%–100% of immatures, depending on the species, were aged correctly by iris colour, but in adults the accuracy of ageing by this trait was 62.5%–100% in different species (Table 1). If a bird was aged first according to other traits, the chance was 1.84 times greater that the iris would be judged to be intermediate than if it was aged by the iris colour first ($\beta_{\text{Identification method [othertraits]}} = 0.610 \pm 0.248$ SE, $\chi^2 = 6.043, p = 0.014$). The effect of species on the probability of the iris colour meeting expectations was not significant ($\chi^2 = 8.956, p = 0.346$), neither were the effects of the body mass index (BMI, $\chi^2 = 2.157, p = 0.142$) or sex ($\chi^2 = 0.133, p = 0.715$).

## DISCUSSION

Our results show that iris colour is a reliable character for determining age in a large set of passerine species on autumn migration, especially for immatures (almost 98% consistency). We suggest the use of the iris colour particularly in species in which ageing by other criteria is not straightforward, as the Long-tailed Tit *Aegithalos caudatus* and Eurasian

Treecreper *Certhia familiaris* (Table 1). In other species it can be used as an additional supporting ageing criterion, in concordance with the results of the studies from Falsterbo Bird Observatory, on a different set of species than we examined (e.g., *Karlsson, Persson & Walinder, 1988*; *Karlsson, 2016*). As the consistency between this trait and the age of an individual was greater when age was determined first by iris colour and then by plumage (and not the other way around), ageing by the iris colour seems to be a good criterion when applied first and intuitively, but can undergo an unconscious bias when used after other criteria. Thus, we suggest that the ringers using the iris colour identify it first, before examination of plumage.

Most adults were aged correctly by iris colour, but the proportion of adults that did not match the typical colour and had the intermediate or even immature (2 individuals) type of iris was larger than among immatures in all species (Table 1). Perhaps the progression towards the brownish hue typical for adults is subject to individual variation; for some non-passerines this process takes longer than a year (*Newton & Marquiss, 1982*; *Karlsson, Persson & Walinder, 1988*; *Bortolotti, Smits & Bird, 2003*). This could be supported by the cases of two adult birds in our study showing an intermediate iris colour, one Robin and one Chaffinch *Fringilla coelebs*. Both had been ringed locally in spring 2019 as second calendar year birds. During our study in autumn 2019 they would have been *ca.* 1.5 years old (*Nowakowski, Muś & Stępniewski, 2012*). Though these individuals had already attained their adult plumage after a complete summer moult and did not differ from older individuals by plumage, their iris colour had still not attained its proper adult tinge. This suggests that iris colour might be used to distinguish individuals in the second calendar year of life among birds in adult plumage, which has been shown in some species (e.g., Eurasian Reed Warbler or Willow Warbler *Phylloscopus trochilus*; *Karlsson, Persson & Walinder, 1988*; *Karlsson, 2016*). This characteristic should be confirmed for each species during moult, when it is still possible to identify these individuals by plumage. However, in the Great Reed Warbler *Acrocephalus arundinaceus* adult birds of known age changed their eye colour between recaptures both ways, from immature to adult type and vice versa, with similar frequency (*Mero & Zuljevic, 2015*). This phenomenon awaits further study, possibly with a set of recaptured adult birds whose precise age is known.

We did not find any relationship between sex and iris colour. In European passerines sexual dimorphism in this trait has been described for only a few species, such as the European Starling *Sturnus vulgaris* (*Smith et al., 2005*). Sexual hormones might influence iris colour in some species of birds (*Trauger, 1974*; *Feare et al., 2015*), as well as the timing when the adult type of iris is attained in males and females (*Newton & Marquiss, 1982*; *Rosenfield & Bielefeldt, 1997*). Our results suggest that iris colour is not related to sex in the species we studied, which could be sexed by plumage. This should be tested by further research in the breeding season, when sexual hormone activity is more pronounced. Similarly, we did not find any relation between the colour of irises and the physiological condition of birds. Thus it seems that this trait is not influenced by metabolic pathways during migration and therefore it can be used as a reliable age criterion, unaffected by the bird's physical state, which changes rapidly during migration.

## CONCLUSIONS

Our study demonstrates that the variation of iris colour in passerines in late autumn is strongly age-related and unaffected by sex or the condition of an individual. Iris colour can therefore be recommended as an additional ageing criterion for a larger set of species than previously described. This method would be particularly useful for ageing species with only slight plumage differences between juveniles and adults, as well as for species that undergo a complete post-juvenile moult and do not display differences between age classes. The Long-tailed Tit is a good example of this category; but the technique would require further testing on a sample of adults during moult. This method has proven more reliable for immatures than adults. Birds with an intermediate iris colour should not be aged upon this character alone. This method does require experience in recognizing the categories of iris colour and we recommend testing the method by comparison with other traits and with the help of an experienced ringer. Potential variations of iris colour in other stages of a bird's life cycle, as well as any influence of other factors, await further studies.

## ACKNOWLEDGEMENTS

Monika Broniszewska, Joanna Rogozińska, Jacek Rogoziński and other volunteers at the Bukowo ringing station made a considerable contribution to this study. Monika Broniszewska and Anna Woźnicka helped to create the database for this study. Tomasz Cofta, Artur Goławski, Łukasz Jankowiak, Jarosław K. Nowakowski, Adrian Surmacki and Liliana Sadzik helped to improve this manuscript. Michael Wink, Jonas Waldenström and Jochen Dierschke provided useful comments on the earlier version of the paper. Joel Avni commented on and edited earlier drafts.

### Funding

The fieldwork and the collation of databases for this study were supported by a Special Research Facility grant (SPUB) from the Polish Ministry of Science and Higher Education to the Bird Migration Research Station, University of Gdańsk (38/E-335/SPUB/SP/2019). The funders had no role in study design, data collection and analysis, decision to publish, or preparation of the manuscript.

### Grant Disclosures

The following grant information was disclosed by the authors:
Special Research Facility grant (SPUB) from the Polish Ministry of Science and Higher Education to the Bird Migration Research Station.
University of Gdańsk (38/E-335/SPUB/SP/2019).

### Competing Interests

The authors declare there are no competing interests.

## Author Contributions

- Michał Polakowski conceived and designed the experiments, performed the experiments, prepared figures and/or tables, authored or reviewed drafts of the paper, and approved the final draft.
- Krzysztof Stępniewski conceived and designed the experiments, prepared figures and/or tables, authored or reviewed drafts of the paper, and approved the final draft.
- Joanna Śliwa-Dominiak and Magdalena Remisiewicz analyzed the data, prepared figures and/or tables, authored or reviewed drafts of the paper, and approved the final draft.

## Animal Ethics

The following information was supplied relating to ethical approvals (i.e., approving body and any reference numbers):

All the bird ringing was conducted with the approval of the General Directorate for Environmental Protection in Poland (DZP-WG.6401.03.2.2018.jro).

## Field Study Permissions

The following information was supplied relating to field study approvals (i.e., approving body and any reference numbers):

Field experiments were made to order of Bird Migration Research Station, Faculty of Biology, University of Gdańsk, Gdańsk, Poland. Permission to access the study site was granted by the Marine Office in Słupsk. Field research at Bukowo was approved by the Marine Office in Słupsk (OW-A-510/100/19jt). Head of the Marine Office in Słupsk who signed a permit was kpt. ż. w. Włodzimierz Kotuniak.

## Data Availability

The raw data and R Code are available as Supplemental Files.

## Supplemental Information

Supplemental information for this article can be found online at http://dx.doi.org/10.7717/peerj.9188#supplemental-information.

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
