# Peer review of "Age-dependent differences in iris colouration of passerines during autumn migration in Central Europe"

_PeerJ, doi:10.7717/peerj.9188_

## Round 0.1 · original submission · Major Revisions

Dear authors
your ms has been reviewed and the reviewers request a thorough revision.
Kind regards
M. Wink
AE

·

Basic reporting

Capturing birds for ringing purposes is common technique for both amateur and professional ornithologists. While species identity is often easy, ageing and sexing birds in the hand is more difficult, and requires training. Criteria used are most often based on slight differences in colouration and wear between juvenile, post-juvenile or adult feather; hence, an understanding of moult patterns is a key feature for plumage-based ageing and sexing. In this paper, the authors expand on eye coloration differences in dark-eyed passerines as an additional means of ageing.

Eye-colouration as an age criterion is already used for some species with specific eye colors/patterns that changes between juvenile and older birds (such as e.g. barred warbler, golden oriole, lesser whitethroat), but a more general use across passerines is not widely used. To my knowledge, Falsterbo Bird Observatory in Sweden is one of few sites that have used it routinely on a broader set of species. Although written mainly in Swedish journals, I think some of the papers from Falsterbo could be useful for the current study. For data on willow warbler: https://www.falsterbofagelstation.se/arkiv/pdf/304.pdf; whinchat https://www.falsterbofagelstation.se/arkiv/pdf/158.pdf; reed warbler https://www.falsterbofagelstation.se/arkiv/pdf/125.pdf; robin https://www.falsterbofagelstation.se/arkiv/pdf/110.pdf; reed bunting and common whitethroat https://www.falsterbofagelstation.se/arkiv/pdf/111.pdf; tree pipit https://www.falsterbofagelstation.se/arkiv/pdf/157.pdf. These papers generally agree with the premise of the current study: that eye color changes with age and could be an additional criterion for ageing birds in the hand.

In evaluating this manuscript, I was asked to first comment on basic reporting, and then move on to validity of findings and experimental design.

The first of the base reporting criteria relates to language, and in my opinion, your text reads fairly well and it is easy to understand – in most parts – what you have done, and why. However, there are a number of smaller grammatical errors in the text, as well as some scope for increasing clarity, that would render the paper easier to read. For instance, in the text you have not settled on one term for what you study, but use ‘eye colour, iris colour, colour of eyes, colour of irides’. A general overhaul of the text should help to identify these cases; I have not had time to scrutinize language in detail.

The second point relates to context of the paper in relation to the research field as a whole, i.e. how well introduction and background reflects what is known. I am not up to date on all literature on eye colours, but the introduction feels adequate in terms of setting the scene and discussing what has been done previously. With that said, I think the examples from Falsterbo mentioned above should be considered to include for both introduction and discussion.

Of the last basic reporting points, I would like to see a better use of figures. Figures 1-3 of goldcrest eyes are not easy to compare side by side, as the birds are in different angles and the focus on the eyes differs, with some interference of the flash. To a non-trained ornithologist, the difference between the birds in current Figure 3 is not easy to see. High quality, side-by-side picture of the eyes would be a good thing to add for a collection of representative species. If possible, all pictures of species/individuals could be presented as extra material online. In terms of raw data, the authors presently supply an excel file with data. The data is readable, but could be improved by clarifying what the different codes represent. For instance, species names could be written in full (rather than having to guess what turmel is) and you could describe what the values in columns E and K represent.

Experimental design

My comments on experimental design is largely connected to validity of findings, please see under the next heading for a discussion on this. For the base premises of experimental design, I believe that manuscript adheres to the Aims and Scopes of the journal, that research question is defined, but that some additional analyses are needed in order to strengthen the paper (please see below).

Validity of the findings

My main concern is to what extent the current data can be used to suggest that eye colour is a ‘reliable feature for age determination in field studies’ (as you say in the abstract) / ‘reliable mean of age determination to larger set of species than previously described in the literature’(as you say in the conclusions). My first concern with such statements is that it implies that checking the eye colour is on par to plumage-based ageing, rather than an additional criterion. For the species that you list in Table 1, the ageing of birds on plumage criteria is fairly straightforward. For instance, ageing a blackbird as juvenile or adult in autumn is easy on plumage (retained juvenile great coverts, primary coverts, pointed retrices etc), and I don’t see how eye colour would improve this determination significantly in practice. A similar argument could be made for other species in the list.

Second, for the data provided in Table 1, the distribution of class 1 and class 2 largely, but not completely, follows plumage-based ageing, while class 3 is more a mixed bag. Your statistics says the method works well for juveniles, but only in 70% for adult; thus for adults, this method is not really that reliable. My third concern is whether classification of eye colour in the paper can be uncoupled from plumage-based ageing. My understanding is that one person both aged the birds based on plumage (and skull ossification) and then classified eye colour of aged birds. Unfortunately, this may increase a risk of introducing bias in the data. That is, if a bird has been classified as adult on plumage it may be more likely to get a classification as adult also on the eye colour because of unconscious bias. I would have liked to see how other people would classify different eyes. That is, 1) one person ages the birds according to plumage criteria, 2) takes pictures of the eyes of all birds, and 3) present these photos to others to classify in a blinded way. In such a setting it would be easier to get a measure of how well the two ways of ageing holds up. At present, your hypothesis is very likely correct, but the setup used can not really separate from other hypotheses. If you have pictures, it would be quite easy to test, and I would urge you to do so. It would also be an important resource for other ringers if those pictures and classification was accessible on the web.

Additional comments

I appreciate the opportunity to read this paper, and I hope you find the comments above helpful in revising it. Eye colour as ageing criterion is something that has often been discussed when I have been part in ringing activities, and a thorough guide on how to do this is warranted in the community.

·

Basic reporting

The English is generally good. However, especially in the abstract there are some phrases that I would correct (e.g. in the northern Poland => in northern Poland). But then I’m not a native speaker, so I recommend to give the manuscript additionally to a native speaker for corrections.

Experimental design

Excellent

Validity of the findings

Although I’m not a good statistician, the findings are supported by generally accepted tests, so the validity is good.
However, it should be added in the conclusions that ageing of birds with intermediate iris colour is not recommended.

Additional comments

It would be nice to add the variation of percentage on species level. In the text this could be done in general (varies between …%), in table 1 a column for percentage of reliable aged birds in adults and first-years could be added.
In the conclusions it could be added that these results may enable ringers to age safely not only birds difficult to age by plumage (mentioned), but also for those impossible to age by plumage (not mentioned), namely those undergoing a complete postjuvenile moult.

---

## Round 0.2 · accepted · Accept

Dear authors
good news! Your revision is adequate and meets the points raised by the reviewers. Therefore, we can accept your manuscript
Greetings
M. Wink
AE

·

Basic reporting

I am happy to say that this is a very much improved manuscript, and all my comments have been addressed in a satisfying way.

Experimental design

No comment

Validity of the findings

No comment

Additional comments

Again, thanks for the effort made in revising the paper. I think it stands much stronger now, and am happy to endorse it. One thing that puzzled me was that there seemed to be two versions of the abstract in the files sent to me, but I guess it is the correct version in the file that is formatted as a manuscript (and the other probably pasted in at submission portal).